# The Human Thyroid-Derived CI-huThyrEC Cell Line Expresses the Thyrotropin (TSH) Receptor and Thyroglobulin but Lacks Other Essential Characteristics of Thyroid Follicular Cells

**DOI:** 10.3390/biom15030375

**Published:** 2025-03-05

**Authors:** Mathias Halbout, Peter A. Kopp

**Affiliations:** 1Division of Endocrinology, Diabetes and Metabolism, University Hospital of Lausanne, University of Lausanne, Hôtel des Patients, Avenue de la Sallaz 08, CH-1011 Lausanne, Switzerland; 2Faculty of Biology and Medicine, University of Lausanne, 1015 Lausanne, Switzerland

**Keywords:** thyroid, cell line, iodine, sodium iodide symporter, thyroid peroxidase

## Abstract

***Background: ***Thyroid hormone synthesis requires the normal function of thyroid follicular cells and adequate nutritional intake of iodine. For in vitro studies on thyroid cell pathophysiology, the immortalized FRTL5 rat thyroid cell line and a derivative thereof, the PCCL3 cell line, are widely used. However, a permanent human thyroid cell line is currently lacking. A recent report described a cell line obtained from human thyroid cells designated as Cl-huThyrEC. ***Methods***: Four clones of Cl-huThyrEC cells were obtained and cultured in the presence of thyroid stimulating hormone (TSH). The expression of key genes defining the thyroid follicular cell phenotype was determined by reverse-transcription PCR (RT-PCR) in FRTL5, PCCL3, and Cl-huThyrEC cells. The latter were cultured as monolayers and as organoids in Matrigel. Iodide uptake was measured and compared among the cell lines. ***Results:*** Gene expression analysis reveals that Cl-huThyrEC cells express the thyroid-restricted transcription factors (*PAX8*, *NKX2.1*, *FOXE1*), the TSH receptor (*TSHR*), and thyroglobulin (*TG*), but they do not express the sodium-iodide symporter (*NIS*), thyroid peroxidase (*TPO*), and pendrin (*SLC26A4*). In functional studies, Cl-huThyrEC cells are unable to concentrate iodide. ***Conclusions:*** Despite the expression of certain key genes that are limited or restricted to thyroid follicular cells, Cl-huThyrEC cells lack some of the essential characteristics of thyroid follicular cells, in particular, *NIS*. Hence, their utility as a model system for thyroid follicular cells is limited.

## 1. Introduction

The thyroid synthesizes and secretes the two thyroid hormones thyroxine (T4) and triiodothyronine (T3). T4 and T3 contain four and three iodine atoms, respectively. These hormones are essential for normal development, growth, and metabolism. Their synthesis occurs in the thyroid follicle, which is formed by an epithelial monolayer consisting of thyroid follicular cells or thyrocytes [1]. The first step of thyroid hormonogenesis consists of the uptake of iodide into thyroid follicular cells against an electrochemical gradient through the sodium-iodide symporter (NIS/SLC5A5) [2]. After transport into the follicular lumen and oxidation, iodide is then incorporated into selected tyrosyl residues within thyroglobulin. Subsequently, mono- and diiodotyrosines are coupled to form T4 or T3 [1].

Among other model systems, primary thyroid cells and immortalized thyroid cell lines form a useful tool for the study of thyroid physiology and pathology in vitro [3]. Two rat thyroid-derived cell lines, FRTL5 cells and a derivative thereof, PCCL3 cells, were generated in the 1980s and have been used in numerous studies. These cells express NIS and concentrate iodide, and they express the key genes defining thyroid function, including *thyroid peroxidase* (*TPO*) and *thyroglobulin* (*TG*), among others [4,5,6,7]. A permanent thyroid cell line of human origin is not currently available, which is a significant deficiency because there is evidence for species-related differences. A thyroid follicular cell line transformed with SV40 (simian virus 40), designated as Nthy-ori 3-1 cells, has been used as a model system in the past [8]. However, Nthy-ori 301 cells do not express the thyroid stimulating hormone receptor (TSHR) and NIS and are unable to concentrate iodide [9]. Primary human thyroid cell cultures are limited by the fact that they tend to dedifferentiate into fibroblast-like cells after about three weeks [10]. Human thyroid microtissue models and organoids have been developed but are not available for widespread use [11,12,13,14,15,16]. Of particular interest are transplantable murine and human organoids derived from induced pluripotent stem (iPS) cells and embryonic stem (ES) cells, but these very sophisticated and labor-intensive techniques are only available in highly specialized laboratories [11,12,13,14,15,16,17,18,19].

For the reasons outlined above, a recent report on the isolation of four clones of human immortalized thyrocytes, designated as Cl-huThyrEC1-4, has been met with interest [20]. Hopperstad and colleagues demonstrated that the Cl-hyThyrEC1-4 cell lines share morphological and functional features of primary human thyrocytes. Moreover, some of the clones showed a TSH-dependent expression of TG in two-dimensional (2D) and three-dimensional (3D) cultures. Under culture conditions with charcoal-stripped serum, T4 could be measured in the medium, but there was no increase in its synthesis in response to TSH. Aside from the presence of NKX2.1 expression, other hallmarks defining differentiated thyroid follicular cells were not characterized.

The aim of the study presented here was to further characterize the cell line Cl-hyThyrEC1-4 and to investigate the iodide transport system in more detail.

## 2. Materials and Methods

### 2.1. Cell Culture

The four human CI-huThyrEC clones (INS-CI-1017), which have been derived from human thyroid [20], were purchased from inSCREENex (Braunschweig, Germany). The cells were cultured at 37 °C and 5% CO_2_ using the conditions reported by Hopperstad in InScreenEx INS-ME-1017 medium and with InScreenEx INS-ME-1017BS supplements [20,21].

The rat thyroid follicular FRTL5 and PCCL3 cell lines were kindly offered by Professor Roberto Di Lauro (Stazione Zoologica Anton Dohrn, Naples, Italy) and cultured in F-12 Ham medium with bicarbonate and without L-glutamine (N4888, Sigma-Aldrich, Merck KGaA, Darmstadt, Germany), supplemented with 10% fetal calf serum (F7524, Sigma-Aldrich), and containing 100 U/mL penicillin/streptomycin (P4333, Sigma-Aldrich), 0.5 mU/mL bovine TSH (T8931, Sigma-Aldrich), 10 µg/mL insulin (I0516, Sigma-Aldrich), and 3.2 ng/mL hydrocortisone (H0135, Sigma-Aldrich).

### 2.2. Three-Dimensional (3D) Culture

CI-huThyrEC-4 cells (~90,000) within 60 µL of medium were cultured inside 60 µL of Matrigel (Sigma Aldrich CLS356252) in 12-well plates. The Matrigel was solidified by incubation for 30 min at 37 °C and 5% CO_2_. One ml of the medium was added to the well, and the cells were grown for eight days. The medium was changed every 48 h.

### 2.3. RNA Extraction, RT-PCR, and qPCR

Cells were harvested with TRIzol^®^ (Thermo Fisher Scientific, Ecublens, Switzerland), and RNA was extracted according to the manufacturer’s recommendations. Subsequently, 1.2 µg of total RNA was reverse transcribed with RevertAid reverse transcriptase (Thermofisher), RiboLock RNase Inhibitor (Thermofisher), and random hexamers in a total volume of 20 µL. As the negative control, pure water instead of RNA was reverse transcribed. The resulting cDNAs were diluted ten-fold, and 5 µL was used to perform a quantitative PCR with a commercial KAPA SYBR Fast master mix (Sigma Aldrich) and ROX high reference dye in a StepOnePlus™ Real-Time PCR system (Thermofisher) using the delta cycle threshold (Ct) method and, as calibrator genes, *TBP* and *GAPDH* for human cDNA, or *Tbp* and *B-actin* for rat cDNA. Forward and reverse primers were chosen to recognize different exons in order to avoid genomic amplification. The primer sequences are listed in Appendix A. The same volume of pure water was added as the negative control for each RT reaction instead of adding 1.2 µg of RNA. As the positive controls, two human cDNAs from normal human thyroid tissue (a gift of Prof. Stefano La Rosa) were included. All reactions were performed in triplicates. To assess the relative expression of the mRNAs of interest, Ct values were determined and compared to the negative controls that showed undetectable values or high Ct values. One-way ANOVA with the Greenhouse–Geisser correction was used to compare the values obtained from the cDNA samples to the negative controls.

### 2.4. Western Blot

After two washes with cold PBS, cells were harvested using RIPA buffer (50 mM Tris HCl, 150 mM NaCl, 1.0% IGEPAL630, 0.5% Sodium Deoxycholate, 1.0 mM EDTA, and 0.1% SDS at a pH of 7.4) complemented with complete protease inhibitor (Roche, 04693132001. Sigma-Aldrich), 50 mM sodium fluoride, 100 mM sodium orthovanadate, and 200 mM of sodium pyrophosphate. Cell lysates were sonicated and clarified by centrifugation for 30 min at 27,000× *g* at 4 °C. Protein concentrations were determined using the Bradford protein assay. Equal protein amounts were denaturized with Laemmli loading buffer and 5 min incubation at 72 °C and loaded on a gradient 4–12% BIS-TRIS polyacrylamide gel (Invitrogen, NP0321BOX. Thermo Fisher Scientific). Electrophoresis was performed in MOPS buffer (50 mM MOPS, 50 mM Tris, 1 mM EDTA pH 8.0, and 0.1% of SDS). Proteins were transferred onto PVDF membranes using a tank transfer system (Bio-Rad, Cressier, Switzerland), and membranes were blocked with 5% non-fat dry milk in T-TBS buffer (0.5% Tween-20 in 20 mM Tris-base pH 7.2, 150 mM NaCl). Incubation with the primary antibody was performed using T-TBS buffer containing 4% bovine serum albumin overnight at 4 °C. The primary antibodies used were all generated in rats, anti-Nis (gift from Nancy Carrasco, [22]), anti-thyroglobulin (Dako, A0251. Agilent Technologies, Basel, Switzerland), and anti-B-actin (Cell Signaling Technologies 4967S. Danvers, MA, USA), and diluted 1:4000. Horse-radish peroxidase-coupled secondary antibodies against rabbit (Sigma a0545, dilution 1:40,000. Sigma-Aldrich) immunoglobulins were diluted in 5% non-fat dry milk T-TBS solution and incubated for 1 h at room temperature. Washing steps after the first and secondary antibodies were performed with T-TBS solution three times for 5 min at room temperature. Signals were detected using the chemiluminescence reagent (Advansta k-12045-d20. San Jose, CA, USA) and a digital image acquisition system (Azure Biosystems, sapphire FL biomolecular imager. Dublin, CA, USA).

### 2.5. Iodide Uptake

Intracellular iodide was measured by two techniques: via the Sandell–Kolthoff reaction as previously described [23], or via the detection of radioactive ^125^iodide, as previously described [24]. Briefly, PCCL3, FRTL5, and huThyrEC-4 cells were cultured at 200,000 cells/mL in 24-well plates. The next day, the medium was removed, cells were washed once with PBS, and 250 µL of uptake buffer (HBSS + 10 mM Hepes) was added to the cells in the presence or absence of sodium iodide (50 µM). For the radio iodide measurement, Na125I (20 mCi/mmol) was added to the 50 µM of cold NaI and incubated with or without 100 µM of potassium perchlorate to block iodide uptake by NIS. Finally, cells were harvested with 100 µL of pure water. Iodide was measured using the Sandell–Kolthoff reaction and compared to a standard curve or via a Gamma Counter to detect ^125^iodide.

## 3. Results

### 3.1. Expression of Thyroid-Restricted Genes in Monolayer Culture

The four Cl-huThyrEC cell clones were cultured using the same conditions as reported by Hopperstad et al. [20]. In the monolayer culture, the cell morphology was similar compared to the original publication (Figure 1A–D). Gene expression analysis by qPCR demonstrates the presence of the three canonical transcription factors defining thyroid follicular cells, i.e., *PAX8*, *NKX2.1*, and *FOXE1* (Figure 2A–C). In addition, *thyroglobulin* (*TG*) is expressed as previously reported (Figure 2D) [20]. Expression of the *TSH receptor* (*TSHR*) was present in clones #2, #3, and #4 but close to background in clone #1 (Figure 2E). However, the essential thyroidal genes *NIS*, *TPO*, and *SLC26A4* were not detectable in any of the Cl-huThyrEC cell lines under standard culture conditions tested since their signal was not higher than the negative control using water (Figure 2F–H). To validate the primers that target *NIS*, *SLC26A4*, and *TPO*, qPCR was performed with two human thyroid cDNAs obtained from normal tissue as the positive control (Appendix A). The results show high expression of these genes in the two samples from normal human thyroid tissue but absent expression in Cl-huThyrEC clone four and no amplification in the negative control. Hence, among the five genes that have a central role in thyroid hormone synthesis, only *TG* is expressed in the four clones, and *TSHR* expression could be demonstrated in three out of four clones.

Moreover, we harvested a cell extract from the four clones of Cl-huThyrEc cells and compared it to the PCCL3 and FRTL5 cell extracts. We could test whether these cell lines express thyroglobulin and the sodium/iodide symporter (Nis) at the protein level. Thyroglobulin (TG) expression is very high in the PCCL3 and FRTL5 cell lines and detectable in the huThyrEC cell lines (Figure 3). Moreover, among the clones of the huThyrEC cell lines, clone 4 showed a higher expression of TG, which corroborates the results obtained at the mRNA level. In addition, Nis was not detectable in the huThyrEC cell lines but was abundantly expressed in FRTL5 and PCCL3.

### 3.2. Expression of Thyroid-Restricted Genes in Three-Dimensional Culture

Cl-huThyrEC cell lines can form organoids when cultured in Matrigel [20]. This is of considerable interest because follicles form the functional unit for thyroid hormone synthesis. To assess whether gene expression is altered in organoids compared to monolayer cultures, the expression levels of thyroid-restricted genes were determined by qPCR, extracting RNA from the cells obtained from the 3D cultures. The cells were cultured as described by Hopperstad et al. in a medium supplemented with insulin, epidermal growth factor (EGF), transferrin, T3, and TSH at 1 mU/mL [20]. In addition, to assess whether there is a response to a higher dose of TSH, the original medium was supplemented with additional TSH (+1 mU/mL resulting in 2 mU/L) in separate triplicates.

When grown on top of a Matrigel layer [12], Cl-huThyrEC cells undergo self-organization and form organoid-like structures within a few days as previously reported [20]. Despite some nuances in the technical preparation (i.e., inside the Matrigel versus on top of the Matrigel), we also observed organoid-like structures after a few days of cell growth (Figure 4).

After eight days of culture, *PAX8*, *NKX2.1*, and *FOXE1* were all detectable (Figure 5A–C). In line with the report by Hopperstad et al., *TG* expression could also be detected in the 3D culture and after increasing the TSH concentration; *TG* expression increased four-fold (Figure 5D). TSHR expression was detectable and unchanged under conditions with 2 mU/L of TSH (Figure 5E). NIS and SLC26A4 were undetectable under both conditions (Figure 5F,H), and TPO was barely detectable (Figure 5G).

### 3.3. Intracellular Iodide Uptake

The uptake of iodide into thyrocytes is central for the synthesis of thyroid hormones. To independently confirm that Cl-huThyrEC cells have lost NIS expression, iodide uptake was measured in the Cl-huThyrEC-4 cells and compared to the FRTL5 and PCCL3 cell lines, which express Nis and Slc26a4 (Figure 3 and Figure 6A,B) [4,5]. As illustrated in Figure 6C, iodide detection using the Sandell–Kolthoff reaction shows that FRTL5 and PCCL3 concentrate iodide, whereas the Cl-huThyrEC-4 cells show no uptake (linear regression significantly non-zero for FRTL5 and PCCL3 but not for Cl-huThyrEC-4 cells (Appendix A)). In addition, we performed a similar experiment using ^125^iodide and a gamma detector in order to measure the iodide content in a more sensitive manner. Here, we observe that FRTL5 cells accumulate ^125^iodide, and the addition of potassium perclorate, which inhibits NIS transport, drastically reduces the amount of ^125^iodide, confirming the specificity of iodide internalization through NIS (Figure 6D). However, ^125^iodide levels remain low in the huThyrEC-4 cells, regardless of whether NIS is blocked or not. Interestingly, this method allows to detect low amounts of ^125^iodide despite the inactivation of NIS in the huThyrEC-4 cells as well as in the FRTL5 cells that are significantly higher than the background (Figure 6E). This may be caused by the internalization of ^125^iodide through a less specific transporter such as chloride channels [25]. Overall, the inability of the Cl-huThyrEC-4 cells to concentrate a high amount of iodide is explained by the absent expression of NIS (Figure 2, Figure 3 and Figure 5) and indicates that these cells have lost an essential trait defining thyroid follicular cells.

## 4. Discussion

The availability of a permanent human thyroid cell line would be a useful tool for in vitro studies for numerous aspects of thyroid cell physiology and cell biology. Therefore, the report on the isolation of the CI-huThyrEC cell line (available as four clones) appeared to offer a long-awaited tool and to fill an important gap.

The results reported here show that CI-huThyrEC cells do express the three pivotal transcription factors (PAX8, NKX2.1, FOXE1), the TSHR, and TG under the culture condition used in the original report. Moreover, CI-huThyrEC cells appear to form organoids.

However, as succinctly presented here, CI-huThyrEC cells do not express *NIS* under these culture conditions, despite the presence of the three transcription factors considered to be sufficient to define and maintain the thyroid follicular cell phenotype. In addition, CI-huThyrEC-4 cells have lost the ability to concentrate iodide, at least under these culture conditions. Furthermore, in addition to the absence of NIS, the expression of several other crucial genes is absent or, at best, barely detectable. This includes *SLC26A4*, which encodes the apical anion exchanger pendrin that mediates iodide efflux into the follicular lumen, as well as *TPO*, the gene coding for thyroid peroxidase that oxides iodide for incorporation into TG [1].

These observations presented here raise numerous questions of interest. First, it is unclear why NIS is not expressed. Aside from the possibility of modifications at the genomic level, the cues leading to the expression of NIS appear to be very complex [26]. On one hand, they require a stimulation of the cAMP pathway, for example, through TSH as performed here, but also an inhibition of the MAPK pathway [26]. During human development, it is striking that the thyroid-restricted transcription factors PAX8, NKX2.1, and FOXE1 and all genes required for thyroid hormone synthesis are expressed early in development except for NIS [27]. NIS expression only starts in developmental week 11–12, and the molecular mechanisms underlying the onset of its expression at that time point remain currently unknown. After a characterization of the genome of huThyrEC cells, a systematic modification of the culture conditions may, perhaps, provide insights into the underlying phenomena.

Whether or not the elements of the hydrogen-peroxide (H_2_O_2_)-generating DUOX/DUOXA system are expressed and functional in huThyrEC cells remains also to be explored. Finally, although huThyrEC cells appear to form organoid-like structures, it is unclear whether they retain a polarized phenotype and whether they are able to form tight junctions.

## 5. Conclusions

A permanent human thyroid cell line should ideally have the following characteristics: a polarized phenotype, formation of tight junctions, ability to form organoids, presence of the three thyroid-restricted transcription factors, expression of all key genes necessary for thyroid hormone synthesis, concentration of iodide and efflux at the apical membrane, retention of regulation by TSH, and the capacity to secrete T4 and T3.

CI-huThyrEC cells may be of interest in studies addressing TG expression and processing or in understanding the cues for NIS expression. In addition, the expression of the TSHR may offer a human in vitro model to assess high-throughput screening of drugs targeting the TSHR. However, at least under the recommended culture conditions, huThyrEC cells fall short of meeting the aforementioned criteria for a functional human thyroid cell line.

## Figures and Tables

**Figure 1 biomolecules-15-00375-f001:**
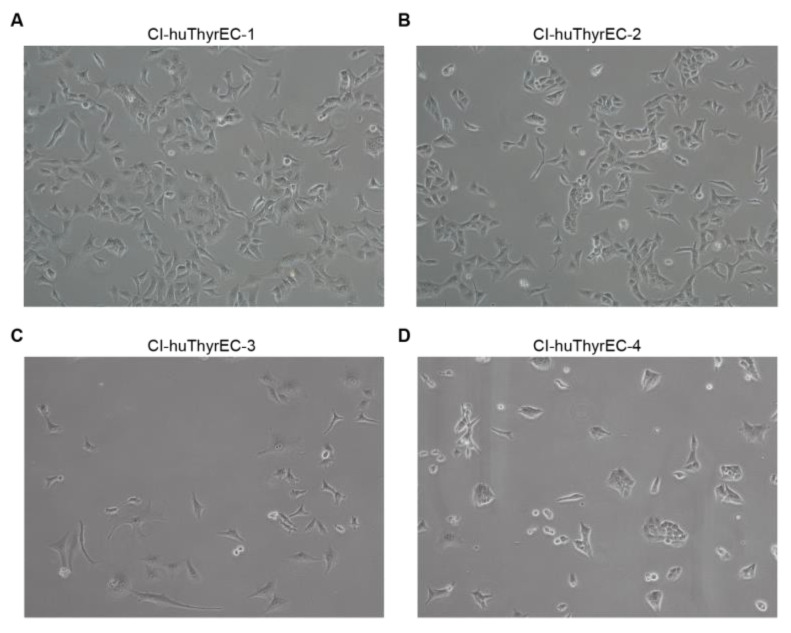
Microscopic appearance of the Cl-huThyrEC cells clones 1 to 4 (**A**–**D**) in the monolayer culture. Magnification 200×.

**Figure 2 biomolecules-15-00375-f002:**
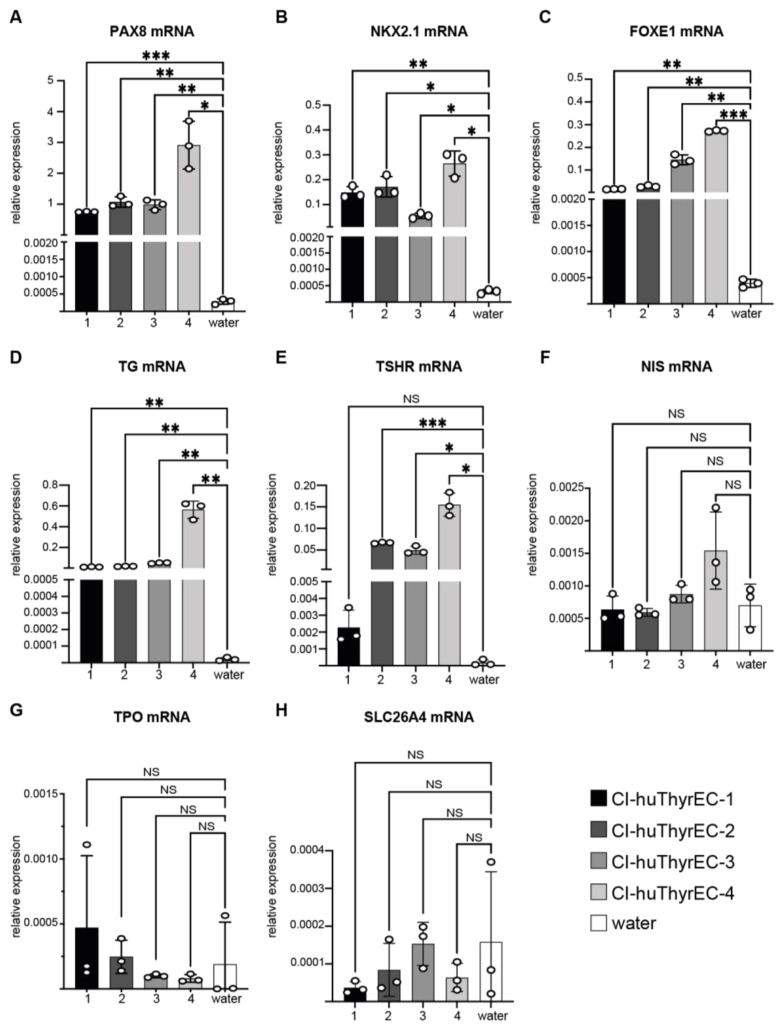
Relative expression of the thyroid-restricted transcription factors *PAX8* (**A**), *NKX2.1* (**B**), and *FOXE1* (**C**) and key genes for thyroid hormone synthesis, *TG* (**D**), *TSHR* (**E**), *NIS* (**F**), *TPO* (**G**), and *SLC26A4* (**H**), in the four CI-huThyrEC clones. For each condition, water was used to determine the background levels. One-way ANOVA with the Greenhouse–Geisser correction was calculated for the cDNA samples and compared to the negative control containing water. Asterisk: *p* < 0.05 = *; *p* < 0.01 = **; *p* < 0.001 = ***. Results show the means of triplicates from one experiment performed independently three times.

**Figure 3 biomolecules-15-00375-f003:**
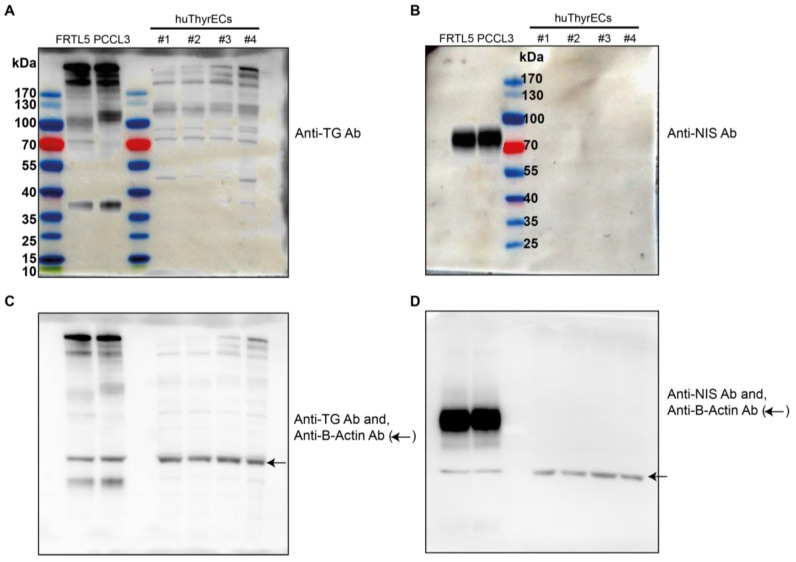
Western blot analysis of FRTL5, PCCL3, and the four clones from Cl-huThyrEC cell lines. Membranes were incubated with antibodies against thyroglobulin (TG, (**A**)) and the sodium iodide symporter (NIS, (**B**)) and merged with molecular weight markers (kilo Dalton, kDa). The membranes were then incubated with an antibody against B-actin (arrow) as a loading control (**C**,**D**). Original images of (**A**–**D**) can be found in Appendix A.

**Figure 4 biomolecules-15-00375-f004:**
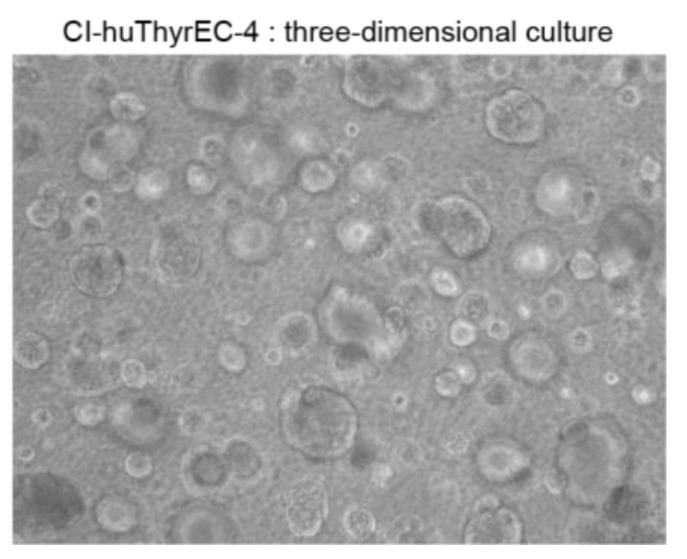
Microscopic appearance of Cl-huThyrEC cell clone 4 cultured in Matrigel. The cells self-organize and form organoid-like structures. Magnification 200×.

**Figure 5 biomolecules-15-00375-f005:**
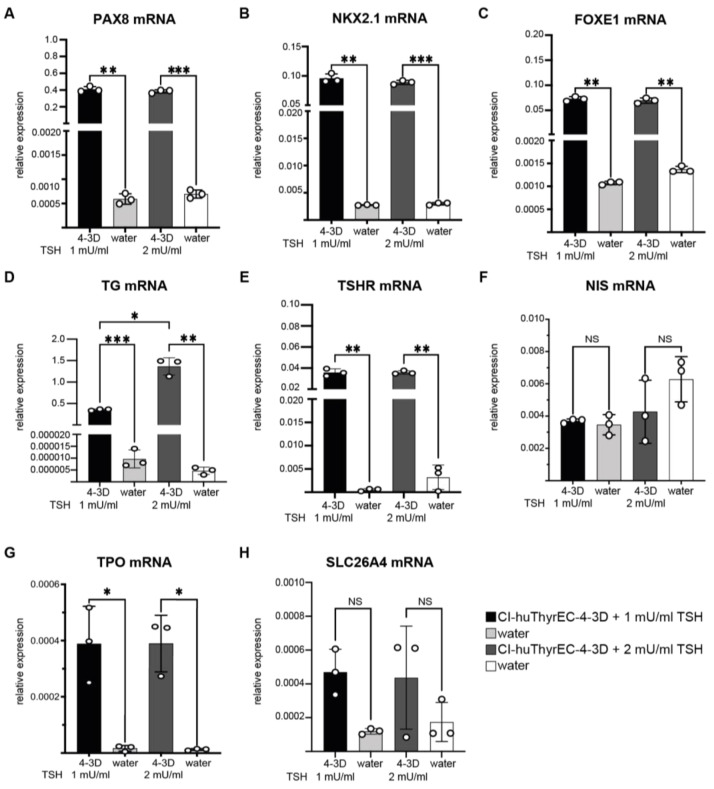
Relative expression of the thyroid-restricted transcription factors PAX8 (**A**), NKX2.1 (**B**), and FOXE1 (**C**) and key genes for thyroid hormone synthesis, TG (**D**), TSHR (**E**), NIS (**F**), TPO (**G**), and SLC26A4 (**H**), in CI-huThyrEC clone #4 cultured after organoid formation and cultured with 1 mU/mL (CI-huThyrEC-4 3D + 1 mU/mL TSH) or 2 mU/mL of TSH (CI-huThyrEC-4 3D + 2 mU/mL TSH). For each condition, water was used to determine the background levels. A one-way ANOVA with the Greenhouse–Geisser correction was calculated to compare the values obtained from cDNA samples to the negative control containing water. Asterisks: *p* < 0.05 = *; *p* < 0.01 = **; *p* < 0.001 = ***. Results show the means of triplicates from one experiment performed independently three times.

**Figure 6 biomolecules-15-00375-f006:**
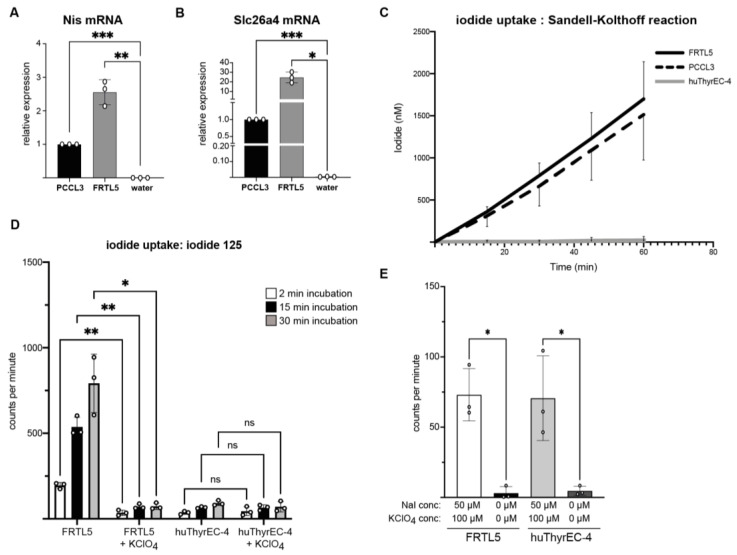
(**A**,**B**) Relative expression of the thyroid-restricted genes *Nis* (**A**) and *Slc26A4* (**B**) in rat PCCL3 and FRTL5 cells. As for the previous experiments, water was used to determine the background levels. A one-way ANOVA with the Greenhouse–Geisser correction was performed to compare the values obtained from cDNA samples to the negative control containing water. Asterisks: *p* < 0.05 = *; *p* < 0.01 = **; *p* < 0.001 = ***. Results show the means of triplicates from one experiment performed independently three times. (**C**) Cumulative intracellular iodide levels in FRTL5 (black), PCCL3 (dotted), and clone 4 of CI-huThyrEC cells (grey) after adding 50 µM of NaI to the medium. NaI levels were measured in triplicates every 15 min via colorimetry using the Sandell–Kolthoff reaction. Bars indicate standard deviations (SD). (**D**) Counts per minute of intracellular ^125^iodide after 2 min, 15 min, or 30 min of incubation with Na^125^I in FRTL5 and huThyrEC-4 cells. A total of 100 µM of potassium perchlorate was added to the incubation buffer in some conditions in order to inhibit NIS transport. (**E**) Counts per minute of intracellular iodide 125 after 30 min of incubation in FRTL5 or huThyrEC-4 cells when incubated with 50 µM of cold iodide (+20 µCi/mmol of Na^125^I) and 100 µM of KClO_4_ or without NaI or KClO_4_ to determine background levels. Asterisks: *p* < 0.05 = *; *p* < 0.01 = **; *p* < 0.001 = ***. Results show the means of triplicates from one experiment. Bars indicate standard deviations (SD).

## Data Availability

The original contributions presented in this study are included in the article/Appendix A. Further inquiries can be directed to the corresponding author/s.

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
