# Peer review of "The Human Thyroid-Derived CI-huThyrEC Cell Line Expresses the Thyrotropin (TSH) Receptor and Thyroglobulin but Lacks Other Essential Characteristics of Thyroid Follicular Cells"

_biomolecules, 2025, doi:10.3390/biom15030375_

Round 1
Reviewer 1 Report
Comments and Suggestions for Authors
The submitted article titled "The human thyroid-derived CI-huThyrEC cell line expresses the thyrotropin (TSH) receptor and thyroglobulin but lacks other essential characteristics of thyroid follicular cells" by Halbout et al discusses the characteristics of a recently published human thyrocyte cell line Cl-huThyrEC, distributed by the company InScreenEx and originally published by Hopperstad et al in 2021.
The topic is interesting and I agree with the authors, that a fully functional human thyrocyte cell line would be of high value - so the in-detail study on a respective available candidate line is indeed a valuable contribution in the field.
While the topic and the overall question on the functionality of the cell line CI-huThyrEC (i.e. four available clones) appears to be important, especially to complement and backup the weak points of the original description, the study itself has some severe drawbacks limiting the conclusions that can be drawn here.
Looking on the title of the study, the authors agree with certain points from the original description (“The human thyroid-derived CI-huThyrEC cell line expresses the thyrotropin (TSH) receptor and thyroglobulin….”), but question its overall status regarding other essential characterisitics, in particular NIS-expression and function.
While the study collects some evidence, models chosen for direct comparison are not very convincing and cell culture conditions appear to be problematic. Same is true for the expression analysis.
Major points
- FRTL5 and PCCL3 cell lines are used as examples for functional thyrocyte cell lines, but they are of rodent origin and also underwent functional changes compared to primary cells – keeping thyrocyte characteristic in monolayer culture is not a feature to be expected from a cell line with physiological behaviour, while it of cause is beneficial in some use cases. To check for a “preserved thyroidal phenotype“ human primary thyrocytes should be used as the performance standard.
- The qPCR experiments for the human genes use “water” as negative control – and no positive control is presented. Are the hNIS primers working properly? What would be the expression level in primary human thyroid tissue under the same culture conditions?
- The study uses the supplied INS-ME-1017 medium + INS-ME-1017BS supplements from InScreenEx – as these products are proprietary is stays unclear, which compounds are included – also in comparison to the FRTL5/PCCL3 culture conditions. As the original publication also uses the hH7 media (with known composition and even better results for some physiological aspects) the use of the InScreenEx products adds a blackbox to the study with no need.
- Also for the iodide uptake results it is hard to decide, what can be expected from a monolayer of “native” primary human thyrocytes. e.g. Kogai et al (2000) report, that “The iodide uptake activity of long-term cultured primary human thyroid cells was relatively low (Fig. 2, control), and the addition of TSH (10 µU/ml, 0·1 mU/ml, or 1 mU/ml) to the basic medium did not further increase the iodide uptake (data not shown).”
Using the Sandell-Kolthoff reaction (that has a very limited ability to detect low transport activities) it can just be assumed, that the CI-huThyrEC have a iodide transport activity not being close to a FRTL5/PCCL3 cell line (if any) – again in monolayer. Kogai et al report in their report, that the long-term cultured primary human thyroid cells had a 12fold induced iodide transport activity by combining follicle formation & TSH stimulation.
- 3D culture conditions appear to differ from the original publication, the current study embeds the single cells into Matrigel, while the original publication refers to a publication of Deisenroth et al on Microtissue culture, where cells are seeded on top of a Matrigel layer
- The authors did not test the reported ability of the CI-huThyrEC to synthesize (very little) thyroid hormone in 3D culture – which would be the integral readout of the cells for an intact (but probably strongly muted) biosynthesis mechanism
In summary, I fully agree, that the report demonstrates limitations of the CI-huThyrEC cell lines, in particular as human test system in the Sandell-Kolthoff-based iodide uptake assay. Further evidence is given, that this function is completely missing, at least under the chosen culture conditions. This evidence could be strengthened by the use of proper positive controls for expression analysis, e.g. human primary thyroid cells or respective tissue samples. I also fully agree, that the cells fail to be a human surrogate for FRTL5 cells – but the study does not demonstrate the lack of the overall capacity of the cell line to become useful under respective culture conditions, e.g. as 3d follicular model – while I would also agree, that this was also not demonstrated by the original publication on these cells to a sufficient degree.
Therefore I would suggest to reject the submission in its current form, but encourage to rephrase certain parts of the paper making clear which use cases cannot be fulfilled by the cell line (as demonstrated datawise) and on the other hand, to discuss weak points of the original publication and raising open questions on this cell line, that would need to be addressed in future studies.
Furthermore I would encourage to strengthen the raised evidence of the experimental results by including respective controls/reference tissue/samples.
Reviewer 2 Report
Comments and Suggestions for Authors
The authors present an investigation into the CI-huThyrEC cell line, a human immortalized thyroid epithelial cell line. The study aims to evaluate its utility as a model for thyroid follicular cells by analyzing gene expression, functionality, and structural characteristics in both 2D and 3D cultures. While the cell line demonstrates some thyroid-specific features, including the expression of key transcription factors (PAX8, NKX2.1, FOXE1), it lacks critical functional components, such as the sodium-iodide symporter (NIS) and thyroid peroxidase (TPO). These deficiencies limit its applicability as a comprehensive model for thyroid hormone synthesis and physiology.
While the study presents intriguing findings, additional improvements are necessary to enhance its impact and meet the standards for publication.
1) The methods are well-documented and reproducible. The inclusion of 3D culture and functional assays adds value. However, investigating alternative culture conditions, such as different TSH concentrations or co-factors, could provide more information and insights into restoring key functional characteristics.
2) The use of Western blot analysis to validate mRNA expression findings, particularly for key markers such as TSHR, TG, and transcription factors, would significantly strengthen the study’s conclusions.
3) The organoid formation is promising, but additional characterization (e.g., tight junction formation, polarity) would enhance the evaluation of these structures.
4) The discussion could be expanded to include potential strategies for improvement. In addition, a more detailed review of previous efforts in developing human thyroid models would strengthen the context. The comparison with established models (e.g., FRTL5) could have been expanded to provide a more comprehensive understanding of the cell line’s relative strengths and weaknesses.
5) I also suggest the authors discuss the potential of CI-huThyrEC cells for specific uses, such as high-throughput screening of drugs targeting TSHR or TG processing.
Round 2
Reviewer 2 Report
Comments and Suggestions for Authors
The Western blot figure is absolutely unclear and cannot be included in the manuscript. The expression of NIS in FRLT5 and PCCL3 is not well-defined, making the gel appear manipulated. Additionally, the figure should include the entire electrophoretic run, without any suspicious cropping or editing. Excessive cropping can compromise the credibility of the results. If only specific bands are shown, it must be clear that the rest of the gel is available and has not been manipulated. It is mandatory to include a molecular weight marker to contextualize the bands and demonstrate that the protein of interest has the expected size.
Comments on the Quality of English LanguageThe quality of English language is acceptable but could be improved to more clearly express the research.
